# Altona Prognostic Index: A New Prognostic Index for ER-Positive and Her2-Negative Breast Cancer of No Special Type

**DOI:** 10.3390/cancers13153799

**Published:** 2021-07-28

**Authors:** Anne-Sophie Wegscheider, Bernhard Ulm, Kay Friedrichs, Christoph Lindner, Axel Niendorf

**Affiliations:** 1MVZ Prof. Dr. med. A. Niendorf Pathologie Hamburg-West GmbH Institut für Histologie, Zytologie und Molekulare Diagnostik, 22767 Hamburg, Germany; AS.Wegscheider@pathologie-hh-west.de; 2Unabhängige Statistische Beratung Bernhard Ulm, 80339 München, Germany; info@usbbu.de; 3Mammazentrum Hamburg MVZ GbR, 20357 Hamburg, Germany; Friedrichs@mammazentrum.eu; 4Agaplesion Diakonieklinikum Hamburg, Frauenklinik, 20259 Hamburg, Germany; Christoph.Linder@d-k-h.de

**Keywords:** breast cancer, unilateral breast cancer, prognosis, progression free survival, histological types of neoplasms

## Abstract

**Simple Summary:**

Breast cancer is the most common tumor-related cause of death in women in Europe and worldwide. The aim of our retrospective study, including 6654 women, was on the one hand to verify the validity of the worldwide known Nottingham prognostic index (NPI), and on the other hand to create a new model with even more prognostic validity. Our newly developed Altona prognostic index (API) shows significantly superior outcome in calculating progression free survival. In contrast to the NPI, the API considers characteristics such as subtypes of breast cancer, as this disease is heterogenous involving different entities, and patient’s age. Evaluating progression free survival in different subgroups, our study shows that both these prognostic indices should only be applied on a patient collective that is ≤70 years old with first primary, unifocal, unilateral breast cancer that is of no special type (NST), estrogen receptor-positive and Her2-negative to get valid prediction data.

**Abstract:**

Breast cancer is a heterogeneous disease representing a number of different histopathologic and molecular types which should be taken into consideration if prognostic or predictive models are to be developed. The aim of the present study was to demonstrate the validity of the long-known Nottingham prognostic index (NPI) in a large retrospective study (*n* = 6654 women with a first primary unilateral and unifocal invasive breast cancer diagnosed and treated between April 1996 and October 2018; median follow-up time of breast cancer cases was 15.5 years [14.9–16.8]) from a single pathological institution. Furthermore, it was intended to develop an even superior risk stratification model considering an additional variable, namely the patient’s age at the time of diagnosis. Heterogeneity of these cases was addressed by focusing on estrogen receptor-positive as well as Her2-negative cases and taking the WHO-defined different tumor types into account. Calculating progression free survival Cox-regression and CART-analysis revealed significantly superior iAUC as well as concordance values in comparison to the NPI based stratification, leading to an alternative, namely the Altona prognostic index (API). The importance of the histopathological tumor type was corroborated by the fact that when calculated separately and in contrast to the most frequent so-called “No Special Type” (NST) carcinomas, neither NPI nor API could show valid prognostic stratification.

## 1. Introduction

The incidence rate of breast cancer among women in Germany was 68,950 in 2016 and 18,600 women died of this disease in 2018. Breast cancer represents the fifth rank of causes of death and is the most common tumor related cause after cardiovascular diseases. These figures correlate with epidemiologic data of breast cancer in Europe as well as worldwide [1,2,3].

Keen interest from the clinical perspective has been turned toward identifying predictive and prognostic markers that will help in predicting prognosis and response to different therapy schemes. The treatment of choice nowadays is breast-conserving therapy with the option of a neoadjuvant or adjuvant chemotherapy, radiotherapy and endocrine therapy. Considering patient’s preferences, decision-making between treatment options can differ, and a subsequent resulting bias of a patient cohort cannot be eliminated. The aim of a therapy plan is a tailored treatment for breast cancer patients, a so-called “personalized medicine” with the best possible outcome for each individual patient [4,5,6,7,8,9]. 

The long known clinicopathological risk factors tumor size, tumor grade and lymph node status, determined by “staging and grading”, have in combination led to the development of a risk prediction model, the Nottingham Prognostic Index (NPI). This index formula divides breast cancer patients depending on the above-mentioned parameters into prognostic groups. For almost 40 years now the NPI is the most popular, commonly used and most often independently validated prognostic index for risk stratification concerning overall survival for breast cancer patients. It has been a benchmark for prognosis assessment all over the world ever since [10,11,12,13,14,15,16].

In the last decades, the NPI has been validated in various and large cohorts. It has been used in its original version with the addition of expression profiles of estrogen and progesterone receptor and Her2 [7,13,17,18,19]. 

The NPI does not consider factors like patient’s age or the histopathological tumor type, which can be divided into “special” and “non-special” types, each of which is defined by the lack of or the presence of characteristic histological and biological features. 

Haybittle et al. calculated with additional variables but none of the factors, including in the model, showed significant increase of the log likelihood, and when incorporated in the index the changes from the results were only marginal [10,11,12,13,20]. 

As biological features, clinical behavior and therapy response vary between these subtypes, representing different entities, it has been difficult to develop accurate risk stratifications based on expected response to different treatment options, metastatic or recurrent disease and overall survival time [21,22,23,24]. 

In the past, it was not possible to fit the information on specific tumor types to therapy schemes. Lack of standardized criteria, low inter-observer reproducibility and relative low prevalence in conjunction with a lack of systematic evaluations are mentionable reasons, and more attention has to be turned on this feature of breast cancer [22].

Independent of these well-known “conventional” parameters, an important amount of research on the evaluation of genetic markers and the use of multigene assays, which are increasingly applied for prognostic stratification and decision support for adjuvant chemotherapy, has been performed in the last two decades. Only a rather homogenous group of patients is included, usually concerning ER-positive, Her2-negative breast cancers, and only some of them define the lymph node status as a relevant criterion. Moreover, most multigene prognostic tests, including Oncotype Dx Recurrence score, Mammaprint, Prosigna (PAM50), are heavily assessing proliferation-related genes, but do not consider typing of breast cancer [25,26,27,28,29,30].

In view of this information, we on the one hand aimed to verify the validity of the NPI using a large cohort of our archives, and furthermore tried to develop a superior model for breast cancer risk stratification. For this purpose, we considered patient’s age at diagnosis as an additional variable. To clean the patient cohort, we filtered these breast cancer cases resulting in more homogeneous groups. Starting with the initial group of 6654 patients, all first primary, unilateral, unifocal and invasive breast cancer cases, we on the one hand applied the NPI and on the other hand developed alternative models as a first step, disregarding additional tumor or patient characteristics. Taking up the above-mentioned heterogeneity of this disease, we then divided the cohort in subgroups. By adding filters concerning different criteria like estrogen receptor- and Her2-status, age and tumor subtype, we got further subdivisions. The intention of this approach was to answer the question of whether these indices depend on the histopathological tumor type.

## 2. Materials and Methods

### 2.1. Patients

For the present analysis 9589 women with a first primary, unilateral, unifocal invasive breast cancer, diagnosed and treated between April 1996 and October 2018 in Hamburg, Germany, were identified and after a positive vote of the ethics commission of the Ärztekammer Hamburg (PV 2946), all patients received several questionnaires. 6654 patients either gave consent or died during this time period and were therefore included in this study. Follow-up questionnaires were collected and edited manually as well as compared and amplified with survival data from the cancer registry. 

### 2.2. Data Management

All data were retrieved from our tumor data bank (MVZ Prof. Dr. med. A. Niendorf Pathologie Hamburg-West GmbH, Institut für Histologie, Zytologie und molekulare Diagnostik). The sample preparation, including processing of the tissue samples, paraffin blocks, staining of HE-slides and immunohistochemical stains for hormone receptor status, etc., was performed using standardized laboratory procedures. Typing and staging of tumors followed rules of the respective WHO and UICC editions [31,32,33,34,35,36,37,38,39]. For the determination of the estrogen receptor status as well as Her2-expression the Remmele and Stegner score, as well as American Society of Clinical Oncology (ASCO)/College of American Pathologists (CAP) guideline criteria were applied [40,41]. Every single case was only signed out after being diagnosed by two independent and experienced board-certified pathologists.

The observation period began after surgery of the invasive breast cancer and ended at the date of death or the last follow-up. The median follow-up time was 15.5 years [14.9–16.8], ending in October 2018.

Recurring (locoregional) disease, diagnosis of distant metastasis and all causes of death were regarded as an event, respectively progression of disease.

Data acquisition was conducted retrospectively and pseudonymized.

### 2.3. Cohort Definition

We conducted the statistical calculations in different subgroups, following distinctive characteristics. At first, all 6654 patients, independent from histological subtype or hormone receptor expression, were included (Table 1). 

We further filtered the data excluding patients with estrogen receptor-negative tumors (*n* = 1088), tumors that were not Her2-negative (*n* = 1159) and patients that were older than 70 years (*n* = 1128) (Figure 1). The age cut at 70 years was determined via survival tree classification (data not shown) and served to exclude the cases with most likely non-tumor-related deaths.

To investigate the impact of the tumor type we separated the 3744 patients again in those subgroups with the diagnosis of breast cancer of no special type, NST (WHO 8500/3, *n* = 2964) which count 79.2% and those with a special type (*n* = 780, all special types, consisting of *n* = 601 only invasive lobular cancers, WHO 8520/3, and *n* = 179 representing all other remaining special types, Table 2).

Our analysis followed a hierarchical approach. We repeated our analysis for each of these different patient subsets, as mentioned above.

### 2.4. Nottingham Prognostic Index

For the initial calculation determining the validity of the NPI and the attempt to create an improved model, we only included patients that gave consent with tumors that were first primary, unifocal, unilateral breast cancers, adding filters for subdivisions following defined characteristics in further steps (criteria mentioned above).

Due to missing data in the category of lymph node status, a separate group “pN missing” was created in order to keep these patients’ data in the analyses. For the calculation of the NPI the missing number of lymph nodes was imputed using linear regression. 

NPI was calculated using
NPI = Histological Grade + 0.2 ∗ Tumor Size + N,(1)
where N was 1 when no positive lymph nodes were examined, 2 when 1–3 positive lymph nodes and 3 if more than 3 lymph nodes were positive [12].

### 2.5. Statistical Methods

Continuous variables were presented using median and range; discrete variables were characterized using absolute and relative numbers.

After each cohort subset collection, the selected subset was split into a training and a testing set using an 80:20 ratio. The training set was used to train the different models. The test set was used for model comparison and evaluation. This step was conducted after each subset selection to achieve evenly distributed cohorts for each subset. 

For progression free survival modeling two different approaches were chosen. Progression free survival was presented using total number of events, if possible, as median progression free survival time with 95% confidence intervals, and if not possible we presented mean progression free survival time with standard deviation. We wanted to include a semiparametric and a nonparametric method and chose the cox proportional hazard regression [42,43] model as semiparametric and survival trees using classification and regression trees (CART) [44] as nonparametric model. We chose the second method for possible nonlinearity of the influencing factors and interactions between those factors as the tree method can handle such circumstances. With the latter we also wanted to examine possible cut points within the variables. As independent variables tumor grade, tumor size, age and nodal status were used.

The predictions of the cox models were classified into risk groups using survival trees. These risk groups were assigned to the test set and shown via Kaplan–Meier plots [45] with log rank tests. As of the nature of classification trees, different numbers of risk groups can be achieved. 

For model comparison area under the time dependent ROC curve (iAUC) and concordance-values were calculated for the developed models. The iAUC is calculated as the area under the curve of area under the receiver operating characteristic (AUROCs) of each event time [46]. The calculations of the event AUROCs and the iAUCs were calculated using the survcomp package [47]. These model performances were benchmarked against the Nottingham Prognostic index. The iAUC values of the cox model, the tree model and the NPI were compared within the test set using Wilcoxon rank sum test for dependent samples. The time dependent AUROCs were plotted for NPI and the API. The API was constructed as the risk score of the cox regression model within the WHO 8500/3 group, a significance level of 5% was defined. Statistical analyses were conducted using R version 4.0.2.

## 3. Results

### 3.1. Total Cohort–6654 Patients

According to our inclusion criteria, the final cohort consisted of 6654 patients. Median progression free survival was 15.5 years [14.9–16.8]. This cohort had no additional filters and consisted of all patients with consent or known death.

All included variables showed a significant correlation with progression free survival (Table 3). The continuous variables age (HR: 1.03 [1.02–1.03]) and tumor size (HR: 1.15 [1.11–1.19]) both increased the risk prediction significantly. The discrete variables pN and histological grade also had significant influence on the progression free survival over time.

The survival tree chose pN as the first splitting variable with the cutoff groups pN0, pN1 and pN1mi to the lesser risk side and pN2a, pN3a and missing N status to the higher risk side (Figure 2). Altogether, 13 different end nodes were developed and every variable was used at least once as a split.

Investigating the whole cohort of patients, our newly developed risk stratification models showed significant superior outcomes compared to the NPI for the iAUCs (Cox (iAUC 0.710; Conc. 0.708) vs. NPI (iAUC 0.639; Conc. 0.668) *p* < 0.001, Tree (iAUC 0.720; Conc. 0.704) vs. NPI *p* < 0.001) (Table 4 and Table 5).

The risk group classification for the cox model provided four groups (Figure 3). The lowest risk groups contained 55.7% of the patient population with an event probability of 9.7%. 7.2% of the test set population was classified as high-risk event probability of 56.7%.

### 3.2. Filtered Cohort–3744 Patients

For a better comparability of cases, we then refined the patient collective for the following analyses and only included patients with primary cancers that were unifocal, estrogen receptor-positive, Her2-negative with an age ≤ 70 years (*n* = 3744), as these factors represent the majority of breast cancer characteristics in our cohort.

Of the 3744 patients 475 had an event, median progression free survival was 12.9 years [15.3-NA].

Emphasizing the age as variable, we see that there are no significant differences between 30- to 70-year-old patients. (HR: 1.00 [0.99–1.02]) (Table 3). The characteristics with the best outcome, according to the CART model, respectively the longest progression survival time were pN0/pN1a/pN1mi/pN2a followed by a tumor size ≤ 3.2 cm and G1 (Figure 2).

In this cohort the cox model provided both the highest iAUC (0.671) and concordance (0. 672). The iAUC of the cox model was significantly better than the iAUC of the tree model and the NPI (both *p* < 0.0001) (Table 5).

### 3.3. Subtype NST (WHO 8500/3)

As 2964 patients from the included 3744 had a diagnosis of “no special type” (NST, WHO 8500/3) and 780 patients a diagnosis of “special type breast cancer” (not WHO 8500/3), statistical calculations for our models were also separately performed in these subgroups.

The median progression free survival time in the collective of NST (WHO 8500/3) could not be calculated because the Kaplan–Meier curve never crossed the 0.5-line, mean survival time was 12.9 years (±0.1) and 373 had an event.

In the cox model in this cohort tumor size, histological type and nodal status showed significant influences on the progression free survival over time. The nodal status showed no significant differences in the influence of pN1a and pN1mi in comparison to pN0. pN2a had about twice the risk and pN3a had about three times the risk of a tumor progression over time in comparison to pN0. Histological grade showed differences in both combinations (I vs. II HR:1.47 [1.07–2.03]; I vs. III. HR: 2.04 [1.38–3.02]). Although age did not show significant influence, it had an additional impact on the model’s performance. The model without age showed a significant worse iAUC (not shown).

The formula of the newly created “Altona prognostic index” (API) was defined as risk prediction of the cox model in this cohort. The formula is
(2)$$\begin{array}{l} Altona\;Prognostic\;Index=\exp(-1.36+0.01*age\;in\;years+0.12*tumor\;size\;in\;cm+\\ \{\begin{array}{c} 0\;\mathrm{if}\;\mathrm{Grade}=1\\ 0.39\;\mathrm{if}\;\mathrm{Grade}=2\\ 0.71\;\mathrm{if}\;\mathrm{Grade}=3 \end{array}+\{\begin{array}{c} 0\;\mathrm{if}\;\mathrm{pN}=0\\ 0.28\;\mathrm{if}\;\mathrm{pN}=1a\\ 0.46\;\mathrm{if}\;\mathrm{pN}=1\mathrm{mi}\\ 0.63\;\mathrm{if}\;\mathrm{pN}=2a\\ 1.07\;\mathrm{if}\;\mathrm{pN}=3a\\ 3.19\;\mathrm{if}\;\mathrm{pN}=X\\ 1.12\;\mathrm{if}\;\mathrm{pN}=\mathrm{missing} \end{array}) \end{array}$$

The iAUC of the API was significantly better than both the NPI and the tree model (iAUC Altona 0.689; iAUC tree 0.642; iAUC NPI 0.664; Cox vs. Tree/Cox vs. NPI/Tree vs. NPI *p* < 0.001). The AUROC values over time show the advantage of the API especially in the prediction between year 1 and 5 (Figure 4).

The API was then divided into three risk groups, where the lowest risk group combined 71% of the test population with a total event rate of 9.5%. The medium risk group contained 26% patients with a total event rate of 25.6%. The highest risk group contained 3% of the test cohort with a total event rate of 60.0%. Although the highest risk group only contained 3% of the test population, it combined 10% of total number of events. The risk group classifications were defined as low risk if the API was ≤ 1.193, medium risk for >1.193 and ≤3.645 and the highest risk group with an API of > 3.645 (Figure 5).

### 3.4. All Special Types (=Not WHO 8500/3)

Highlighting the group of all special types except NST (not WHO 8500/3, *n* = 780, mean progression free survival time was 12.8 years (±0.3), 102 events), it was shown that only the parameters tumor size and lymph node status are significant. API and classification tree analyses compared to NPI did no longer show significant differences.

### 3.5. Subtype Invasive Lobular (=WHO 8520/3)

Dividing these special types again, as the invasive lobular breast cancers (WHO 8520/3) make up the biggest part (*n* = 601, 77%, 12.7 years (±0.3)) of this group, we could no longer create a model-using different statistical methods—that enables a valid risk prediction. Additionally, it could be shown that NPI applied on only this subpopulation does not give applicable information for risk stratification either. There were no significant differences in the compared indices.

Analyzing the subgroup of invasive lobular breast cancers, it was shown again (as seen above in the group of all special types) that the parameter tumor grade is no longer significant, compared to breast cancers of no special type (NST).

There were neither significant differences in training and testing set nor in the different groups with or without consent.

## 4. Discussion

Almost 40 years ago, in 1982, the Nottingham Prognostic Index (NPI) was constructed from a multiple-regression analysis of prognostic factors and survival in a series of 387 patients with primary breast cancer, based on lymph-node stage, tumor size and pathological grade [12]. Since then, the NPI is the most popular, commonly used prognostic index for risk stratification concerning overall survival for breast cancer patients. Other research teams looking at different criteria and pitches could not create an alternative index with significant improvement and validation in Europe or the USA so far [21,48,49]. In New Zealand a specific predictive model for prediction of breast cancer mortality in women with primary invasive breast cancer was developed and showed superior outcome to the NPI in the Auckland and Waikato regions [50].

The objective of our analyses was on the one hand to verify the general validity of the NPI using the data of our large breast cancer tumor data bank, and on the other hand trying to establish a new prognostic model for breast cancer risk stratification with improved prognostic power.

For the validation of the NPI and our Altona prognostic index, named after the district Altona in Hamburg where our institute of pathology is located, we filtered the cohort and divided it into different groups regarding most common features of breast cancer. Our major focus was turned on these clear, filtered subgroups and we wanted to examine if the validity of both indices is still reproducible.

The calculations for both indices started with the total cohort of 6654 patients, including all first primary, unilateral, unifocal invasive breast cancer cases. We compared our predictive model with the NPI under “conventional conditions” as a first step, disregarding additional tumor characteristics. Our API showed significant improvement of predicting progression free survival over time by defining better risk groups. Using the parameter of tumor size as a steady variable by considering continuous tumor size instead of determined groups, extreme cuts could be avoided (i.e., tumor size of 0.9 cm equates to pT1b and size of 1.1 cm equates to pT1c, according to TNM-classification of breast tumors [51]). Patient’s age as an additional parameter in our model is another factor that led to superior outcomes. Although age did not show significant influence as a variable on its own, it had an additional impact on the model’s performance.

In a further step we refined the cohort and only included patients with primary, unifocal, unilateral breast cancer that are not older than 70 years at diagnosis with tumors that are estrogen receptor-positive, Her2-negative and of no special type (NST), as these factors represent the majority of breast cancer characteristics in our cohort. Those patients who are >70 years old show worse prognostic rates as this group of patients have more comorbidities and increased numbers of death not correlating with the diagnosis of breast cancer [52,53,54,55,56]. However, between ≤70-year-old patients the calculations did not reveal significant differences (HR: 1.00 [0.99–1.02]) (Table 3).

The Altona prognostic index arose from the iAUC of this just mentioned cohort showing the model with the best prognostic value. Compared to the NPI, not only the iAUC but also concordance values were superior. Especially in the first five years after diagnosis/initial therapy API shows significantly superior predictive validity.

As breast cancer is such a heterogeneous disease, that involves multiple entities (“special types”) with characteristic histological and biological features, independent from age, hormone receptor status and age, we divided the cohort again concerning these subtypes in additional analyses [22].

601 cases of all 780 special types were classified as invasive lobular carcinomas. Most of the invasive lobular carcinomas are graded intermediate (G2) [57], therefore tumor grade, as one of the four most powerful parameters, is not significant anymore (after excluding the NST). Our results show that neither the NPI nor our prognostic model should be applied in cases diagnosed as the invasive lobular subtype.

The subgroup involving all other special types (excluding NST and invasive lobular carcinomas, Table 1) includes only 179 cases of different entities with various characteristics each. Due to the small number and heterogeneity of cases, neither NPI nor API show valid prediction data.

Considering this finding of significance of the tumor subtype in our index represents a distinctive feature of API, standing out against most of other known indices.

One limitation of this study is the fact that different therapy-schemes were not taken into account since this information was not accessible at the time when this study was performed. We must assume that patients have been treated according to the state-of-the-art standards applicable at the time when their tumors had been diagnosed. Another limitation is the relatively small number of cases especially within the group of patients with the diagnosis of a tumor type other than NST or respectively lobular type carcinomas. However, strict consideration of WHO criteria has led to a comparable prevalence as depicted in our tumor data bank.

In conclusion, our calculations show that these two prognostic indices, NPI and API, are only reliable tools when applied to breast cancer cases, that are of no special type, NST (WHO 8500/3), estrogen receptor-positive, Her2-negative and when affected women are ≤70 years old. Misleadingly, the indices are applicable “overall”, as up to 80% of all breast cancer cases are diagnosed as NST and therefore overshadow the special types by amount [22]. Clinicians and patients should be aware using either NPI or API for individual breast cancer prognosis when a histological subtype other than NST is diagnosed.

Regarding this, in the future research concerning the prognostic criteria to better classify special types of breast cancer, particularly invasive lobular carcinomas, has to be urged as they cannot be compared one-on-one with the largest group, the NST.

Almost 40 years later, still the most important parameters for a good risk stratification are the well-known variables tumor size, tumor grade, lymph node involvement and patient’s age, thus representing irreplaceable clinicopathological basics.

## 5. Conclusions

We created a risk stratification model—namely, Altona prognostic index (API)—that is significantly superior to the well-known and worldwide used Nottingham prognostic index (NPI), concerning all cases that are primary, unifocal breast cancers or regarding those of no special type (NST) that are additionally estrogen receptor-positive and Her2-negative. For the heterogeneous group of the special types, including invasive lobular breast cancers, there still does not exist a valid prediction model.

## Figures and Tables

**Figure 1 cancers-13-03799-f001:**
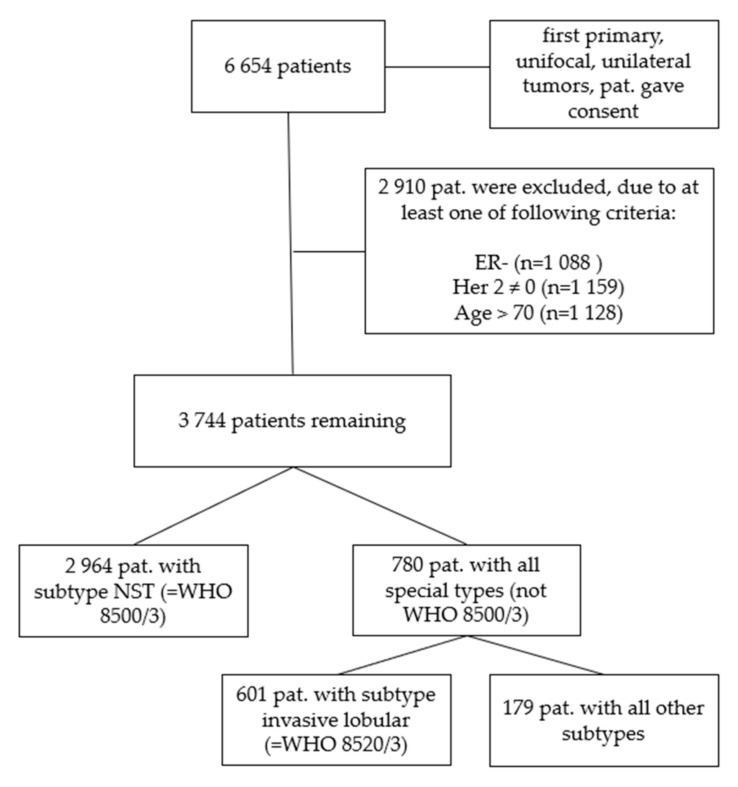
Consort diagram of the patient cohorts. First, all 6654 patients who gave consent with first primary, unifocal and unilateral tumors were analyzed, in step 2 the cohort was subdivided into ER-positive, Her2-negative cases with patients ≤ 70 years of age (*n* = 3744). In a third step only patients with a histological subtype of NST (WHO 8500/3) were evaluated (*n* = 2964). In a fourth step all remaining cases except NST (all other subtypes, not WHO 8500/3) were analyzed (*n* = 780) and in a final step all invasive lobular (WHO 8520/3) tumor cases (*n* = 601) separated from all other remaining subtypes (*n* = 179).

**Figure 2 cancers-13-03799-f002:**
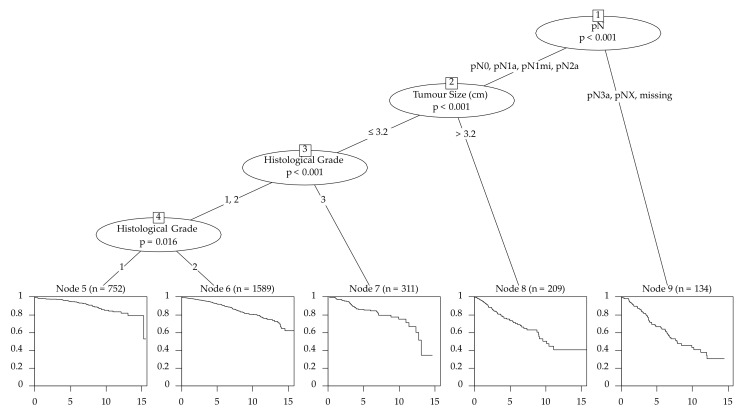
Survival tree of the training set for the filtered cohort–3744 patients (primary cancers that were unifocal, estrogen receptor-positive, Her2-negative with an age ≤ 70 years (*n* = 3744)).

**Figure 3 cancers-13-03799-f003:**
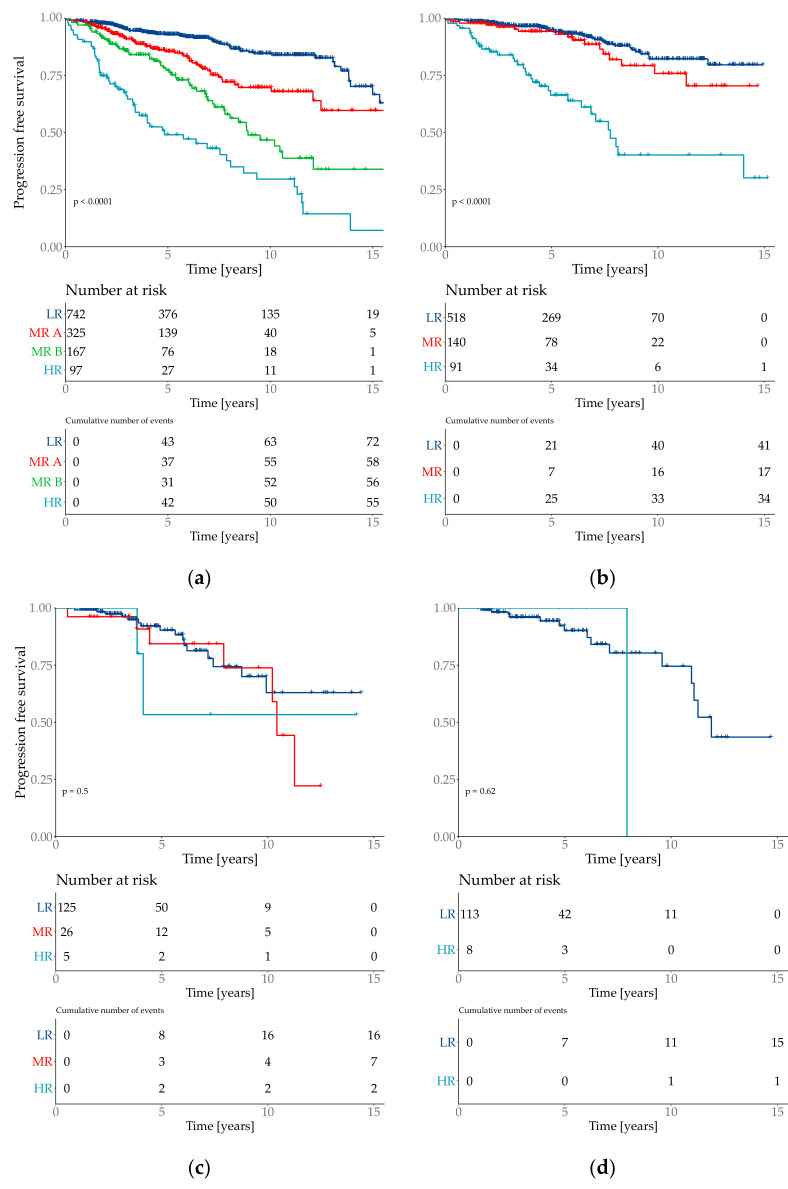
Kaplan–Meier plots of the predicted risk groups. *Y*-axis is progression free survival. Risk groups are specified as low risk (LR), medium risk (MR) and high risk (HR). Plot (**a**) Total cohort–6654 patients with 2 medium risk groups which are specified as MR A and MR B. (**b**) filtered cohort–3744 patients; (**c**) all special types (=not WHO 8500/3); (**d**) subtype invasive lobular (=WHO 8520/3).

**Figure 4 cancers-13-03799-f004:**
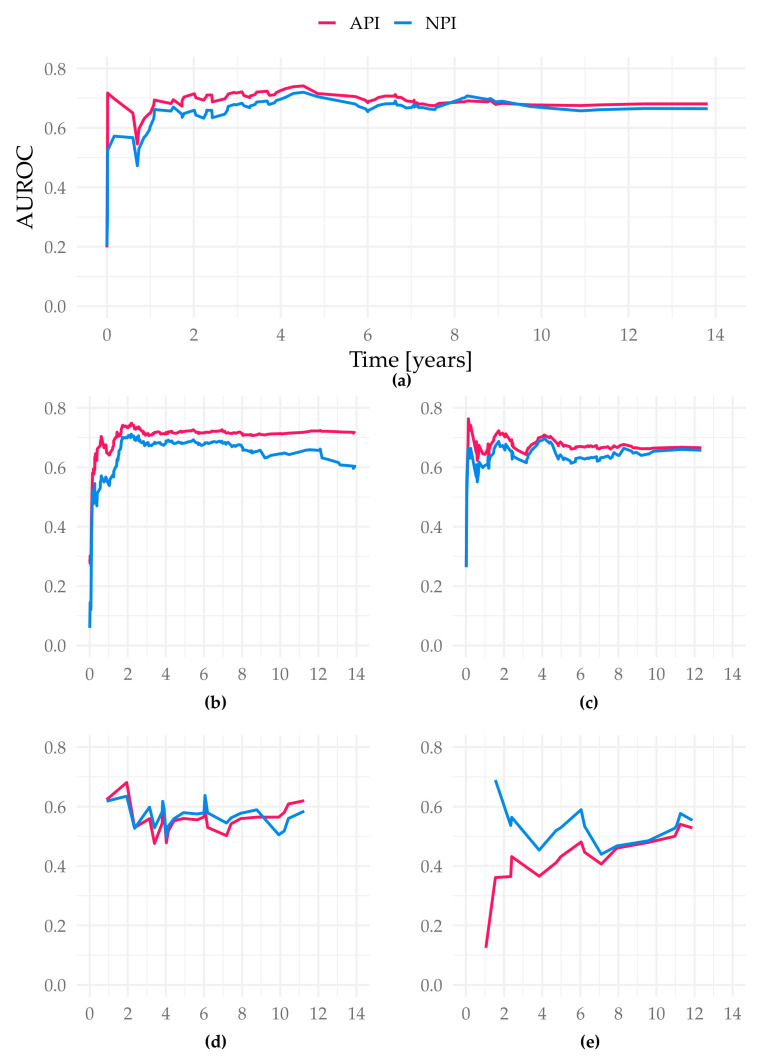
AUROC over time of the different study cohorts in the test sets. Figure (**a**) displays the cohort Subtype NST (=WHO 8500/3), basis for our suggested API; (**b**) Total cohort–6654 patients (gave consent, first primary, unifocal and unilateral tumors); (**c**) filtered cohort–3744 patients (additional filters: only ER-positive, Her2-negative cases, patients ≤ 70 years old); (**d**) all special types (not WHO 8500/3); (**e**) Subtype invasive lobular (=WHO 8520/3); all *X*-axis present time in years. *Y*-axis represent the AUROC values at each event time point.

**Figure 5 cancers-13-03799-f005:**
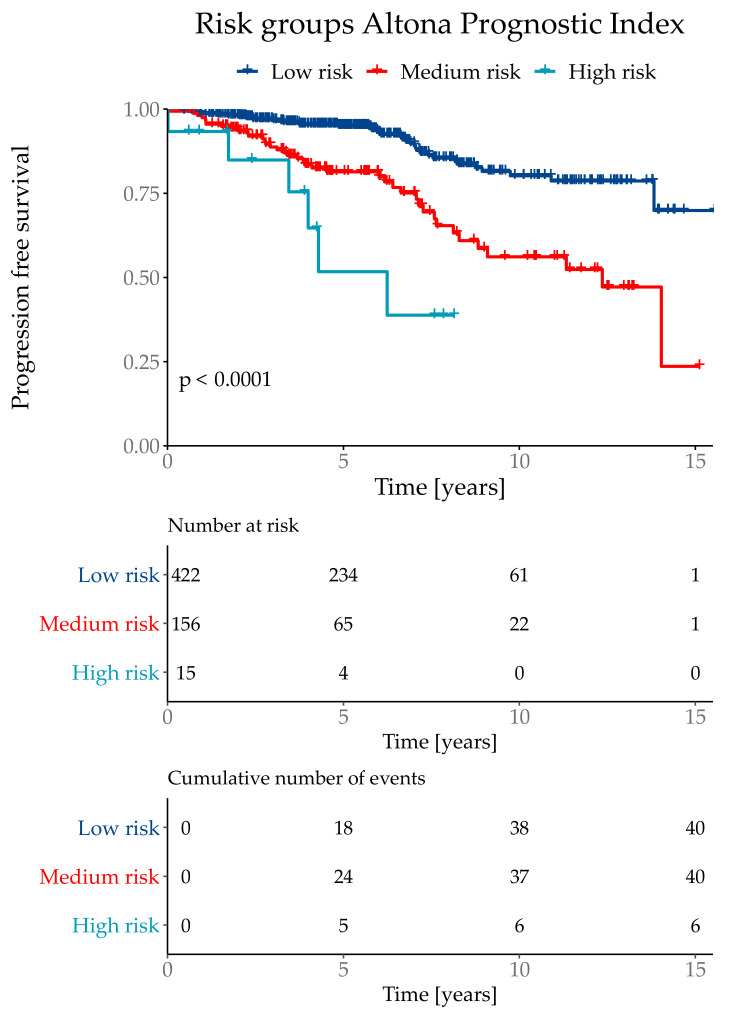
Kaplan–Meier curve with Log-Rank test of the risk classifications in the subtype NST (=WHO 8500/3) cohort.

**Table 1 cancers-13-03799-t001:** Distribution of patients and tumor characteristics among different WHO tumor classifications. Classifications with 10 or less expressions were combined into one category.

WHO Classifications											
	Overall	M 8500/3	M 8520/3	M 8480/3	M 8211/3	M 8507/3	M 8401/3	M 8575/3	M 8500/3, M 8520/3	M 8500/3, M 8480/3	Other
	(N = 6654)	(N = 5394)	(N = 876)	(N = 84)	(N = 81)	(N = 40)	(N = 33)	(N = 30)	(N = 19)	(N = 14)	(N = 83)
Histological grade											
Grade 1	1264 (19.0%)	1115 (20.7%)	17 (1.9%)	36 (42.8%)	81 (100%)	0 (0%)	0 (0%)	0 (0%)	3 (15.8%)	1 (7.1%)	11 (13.3%)
Grade 2	3581 (53.8%)	2615 (48.5%)	783 (89.4%)	46 (54.8%)	0 (0%)	31 (77.5%)	24 (72.7%)	5 (16.7%)	15 (78.9%)	12 (85.8%)	50 (60.2%)
Grade 3	1809 (27.2%)	1664 (30.8%)	76 (8.7%)	2 (2.4%)	0 (0%)	9 (22.5%)	9 (27.3%)	25 (83.3%)	1 (5.3%)	1 (7.1%)	22 (26.5%)
Estrogen receptor (ER)											
negative	1088 (16.4%)	984 (18.2%)	16 (1.8%)	1 (1.2%)	0 (0%)	2 (5.0%)	31 (93.9%)	29 (96.7%)	1 (5.3%)	0 (0%)	24 (28.9%)
positive	5566 (83.6%)	4410 (81.8%)	860 (98.2%)	83 (98.8%)	81 (100%)	38 (95.0%)	2 (6.1%)	1 (3.3%)	18 (94.7%)	14 (100%)	59 (71.1%)
T											
pT1a	266 (4.0%)	226 (4.2%)	23 (2.6%)	1 (1.2%)	15 (18.5%)	0 (0%)	0 (0%)	0 (0%)	0 (0%)	0 (0%)	1 (1.2%)
pT1b	1343 (20.1%)	1111 (20.6%)	151 (17.2%)	8 (9.5%)	43 (53.1%)	13 (32.5%)	2 (6.1%)	3 (10.0%)	4 (21.1%)	0 (0%)	8 (9.6%)
pT1c	2727 (41.0%)	2234 (41.4%)	329 (37.6%)	47 (56.0%)	20 (24.7%)	17 (42.5%)	21 (63.6%)	4 (13.3%)	10 (52.6%)	7 (50.0%)	38 (45.8%)
pT2	2061 (31.0%)	1658 (30.7%)	300 (34.2%)	27 (32.1%)	3 (3.7%)	6 (15.0%)	8 (24.2%)	17 (56.7%)	5 (26.3%)	5 (35.7%)	32 (38.6%)
pT3	253 (3.8%)	161 (3.0%)	73 (8.4%)	1 (1.2%)	0 (0%)	4 (10.0%)	2 (6.1%)	6 (20.0%)	0 (0%)	2 (14.3%)	4 (4.8%)
pT4	4 (0.1%)	4 (0.1%)	0 (0%)	0 (0%)	0 (0%)	0 (0%)	0 (0%)	0 (0%)	0 (0%)	0 (0%)	0 (0%)
Tumor size (cm)											
Mean (SD)	1.9 (1.4)	1.9 (1.3)	2.4 (1.8)	1.9 (0.9)	0.9 (0.5)	2.1 (2.1)	2.0 (1.0)	3.0 (1.7)	1.9 (1.1)	2.9 (1.7)	2.29 (1.38)
Median [Min, Max]	1.6 [0.1, 15.0]	1.5 [0.1, 15.0]	1.8 [0.1, 12.0]	1.8 [0.4, 5.5]	0.8 [0.2, 3.4]	1.5 [0.6, 12.0]	1.7 [0.8, 6.0]	2.8 [1.0, 9.0]	1.5 [0.7, 4.5]	2.1 [1.5, 7.0]	1.90 [0.300, 8.00]
N											
pN0	4454 (66.9%)	3584 (66.4%)	593 (67.7%)	67 (79.7%)	65 (80.2%)	25 (62.5%)	24 (72.7%)	22 (73.3%)	14 (73.7%)	7 (50.0%)	53 (63.9%)
pN1a	1113 (16.7%)	942 (17.5%)	125 (14.3%)	8 (9.5%)	2 (2.5%)	7 (17.5%)	3 (9.1%)	3 (10.0%)	3 (15.7%)	3 (21.5%)	17 (20.5%)
pN1mi	197 (3.0%)	167 (3.1%)	22 (2.5%)	0 (0%)	1 (1.2%)	2 (5.0%)	2 (6.1%)	0 (0%)	0 (0%)	2 (14.3%)	1 (1.2%)
pN2a	430 (6.5%)	350 (6.5%)	62 (7.1%)	5 (6.0%)	0 (0%)	2 (5.0%)	1 (3.0%)	4 (13.4%)	1 (5.3%)	1 (7.1%)	4 (4.8%)
pN3a	258 (3.9%)	197 (3.7%)	52 (5.9%)	0 (0%)	1 (1.2%)	3 (7.5%)	2 (6.1%)	0 (0%)	0 (0%)	1 (7.1%)	2 (2.4%)
pNX	59 (0.9%)	44 (0.8%)	8 (0.9%)	1 (1.2%)	2 (2.5%)	1 (2.5%)	0 (0%)	1 (3.3%)	1 (5.3%)	0 (0%)	1 (1.2%)
Missing	143 (2.1%)	110 (2.0%)	14 (1.6%)	3 (3.6%)	10 (12.4%)	0 (0%)	1 (3.0%)	0 (0%)	0 (0%)	0 (0%)	5 (6.0%)
Age at diagnosis											
Mean (SD)	60.6 (11.9)	60.0 (12.0)	62.8 (10.8)	65.1 (12.8)	61.5 (9.02)	66.7 (8.23)	63.1 (11.0)	65.5 (15.7)	61.2 (13.0)	60.6 (12.2)	65.1 (12.6)
Median [Min, Max]	61 [25, 95]	61 [25, 95]	63 [33, 94]	65 [37, 95]	61 [41, 83]	67 [39, 83]	61 [37, 86]	65 [36, 92]	60 [40, 89]	62 [42, 77]	66 [38, 95]
Human epidermal growth factor receptor 2 (HER2)											
unknown	212 (3.2%)	195 (3.6%)	7 (0.7%)	3 (3.5%)	0 (0%)	1 (2.5%)	4 (12.1%)	0 (0%)	0 (0%)	0 (0%)	2 (2.4%)
negative	5495 (82.6%)	4353 (80.7%)	802 (91.6%)	75 (89.3%)	76 (93.8%)	32 (80.0%)	23 (69.7%)	28 (93.3%)	19 (100%)	12 (85.8%)	75 (90.4%)
positive	587 (8.8%)	545 (10.1%)	19 (2.2%)	3 (3.6%)	1 (1.2%)	7 (17.5%)	6 (18.2%)	2 (6.7%)	0 (0%)	1 (7.1%)	3 (3.6%)
Missing	360 (5.4%)	301 (5.6%)	48 (5.5%)	3 (3.6%)	4 (5.0%)	0 (0%)	0 (0%)	0 (0%)	0 (0%)	1 (7.1%)	3 (3.6%)
Nottingham Prognostic Index											
Mean (SD)	3.90 (1.21)	3.91 (1.24)	3.98 (1.03)	3.23 (0.88)	2.39 (0.51)	4.11 (1.22)	4.03 (0.83)	4.80 (1.02)	3.53 (0.95)	4.23 (0.986)	4.04 (1.05)
Median [Min, Max]	3.50 [2.02, 9.00]	3.52 [2.02, 9.00]	3.46 [2.10, 7.90]	3.25 [2.08, 6.40]	2.20 [2.04, 4.50]	3.44 [3.12, 8.40]	4.20 [3.16, 6.26]	4.56 [3.20, 7.04]	3.30 [2.14, 5.90]	4.47 [2.34, 5.40]	4.12 [2.18, 6.60]

**Table 2 cancers-13-03799-t002:** Distribution of patients and tumor characteristics among unifocal invasive breast cancer cases with additional filters (ER+, Her2-, ≤70) and different WHO tumor classifications (=subtypes). Classifications with 10 or less expressions were combined into one category (“other”).

	Overall	M 8500/3	M 8520/3	M 8211/3	M 8480/3	M 8507/3	M 8500/3. M 8520/3	Other
	(N = 3744)	(N = 2964)	(N = 601)	(N = 62)	(N = 47)	(N = 16)	(N = 13)	(N = 41)
Histological grade								
Grade 1	964 (25.8%)	865 (29.2%)	13 (2.2%)	62 (100%)	16 (34.1%)	0 (0%)	2 (15.4%)	6 (14.6%)
Grade 2	2262 (60.4%)	1628 (54.9%)	550 (91.5%)	0 (0%)	30 (63.8%)	16 (100%)	10 (76.9%)	28 (68.3%)
Grade 3	518 (13.8%)	471 (15.9%)	38 (6.3%)	0 (0%)	1 (2.1%)	0 (0%)	1 (7.7%)	7 (17.1%)
T								
pT1a	163 (4.4%)	135 (4.6%)	17 (2.8%)	11 (17.8%)	0 (0%)	0 (0%)	0 (0%)	0 (0%)
pT1b	926 (24.7%)	752 (25.4%)	121 (20.2%)	31 (50%)	5 (10.6%)	8 (50%)	3 (23.1%)	6 (14.6%)
pT1c	1607 (42.9%)	1304 (44%)	223 (37.1%)	18 (29%)	29 (61.7%)	4 (25%)	7 (53.8%)	22 (53.7%)
pT2	955 (25.5%)	724 (24.4%)	199 (33.1%)	2 (3.2%)	13 (27.7%)	3 (18.8%)	3 (23.1%)	11 (26.8%)
pT3	92 (2.5%)	48 (1.6%)	41 (6.8%)	0 (0%)	0 (0%)	1 (6.2%)	0 (0%)	2 (4.9%)
pT4	1 (0%)	1 (0%)	0 (0%)	0 (0%)	0 (0%)	0 (0%)	0 (0%)	0 (0%)
Tumor size (cm)								
Mean (SD)	1.8 (1.2)	1.7 (1.1)	2.2 (1.7)	0.9 (0.5)	1.8 (0.8)	1.6 (1.5)	1.6 (0. 8)	2.05 (1.33)
Median [Min, Max]	1.5 [0.1, 15.0]	1.4 [0.1, 15.0]	1.7 [0.1, 10.0]	0.9 [0.2, 3.4]	1.8 [0.6, 4.2]	1.2 [0.6, 7.0]	1.5 [0.8, 3.2]	1.6 [0.6, 7.0]
N								
pN0	2635 (70.4%)	2079 (70.1%)	423 (70.4%)	51 (82.3%)	37 (78.7%)	10 (62.5%)	10 (76.9%)	25 (61%)
pN1a	610 (16.3%)	500 (16.9%)	86 (14.3%)	1 (1.6%)	7 (14.9%)	3 (18.8%)	2 (15.4%)	11 (26.8%)
pN1mi	124 (3.3%)	104 (3.5%)	16 (2.7%)	0 (0%)	0 (0%)	2 (12.5%)	0 (0%)	2 (4.9%)
pN2a	203 (5.4%)	158 (5.3%)	41 (6.8%)	0 (0%)	3 (6.4%)	0 (0%)	0 (0%)	1 (2.4%)
pN3a	109 (2.9%)	77 (2.6%)	29 (4.8%)	1 (1.6%)	0 (0%)	0 (0%)	0 (0%)	2 (4.9%)
pNX	13 (0.3%)	7 (0.3%)	2 (0.3%)	2 (3.2%)	0 (0%)	1 (6.2%)	1 (7.7%)	0 (0%)
Missing	50 (1.4%)	39 (1.3%)	4 (0.7%)	7 (11.3%)	0 (0%)	0 (0%)	0 (0%)	0 (0%)
Age at diagnosis								
Mean (SD)	57.2 (9.0)	56.8 (9.2)	58.8 (8.1)	58.3 (6.8)	58.3 (8.4)	61.2 (7.7)	56.2 (8.2)	58.8 (8.3)
Median [Min, Max]	59 [25, 70]	58 [25, 70]	60 [33, 70]	59 [41, 70]	60 [37, 70]	65 [39, 69]	57 [40, 68]	60.0 [42, 70]
Nottingham Prognostic Index								
Mean (SD)	3.61 (1.14)	3.58 (1.16)	3.89 (0.96)	2.38 (0.53)	3.32 (0.91)	3.64 (0.56)	3.39 (0.73)	3.90 (1.00)
Median [Min, Max]	3.30 [2.02, 9.00]	3.30 [2.02, 9.00]	3.44 [2.10, 7.60]	2.20 [2.04, 4.50]	3.30 [2.12, 6.40]	3.37 [3.12, 4.46]	3.30 [2.22, 4.64]	3.90 [2.18, 6.40]

**Table 3 cancers-13-03799-t003:** Results of the multivariate Cox regression models in the different subsets. Hazard Ratios (HR) with 95% confidence intervals (CI) and *p*-values (*p*) are given for each of the 5 different patient cohorts.

	Total Cohort-6654		Filtered Cohort-3744		Subtype NST (WHO 8500/3)		All Special Types (Not WHO 8500/3)		Subtype Invasive Lobular (WHO 8520/3)	
	HR [CI]	*p*	HR [CI]	*p*	HR [CI]	*p*	HR [CI]	*p*	HR [CI]	*p*
Age at diagnosis	1.03 [1.02–1.03]	<0.001	1.00 [0.99–1.02]	0.441	1.01 [1.00–1.02]	0.2	1.00 [0.98–1.03]	0.749	0.99 [0.96–1.02]	0.584
Tumor size (cm)	1.15 [1.11–1.19]	<0.001	1.21 [1.13–1.30]	<0.001	1.22 [1.12–1.33]	<0.001	1.20 [1.06–1.37]	0.004	1.15 [1.00–1.32]	0.056
Histological grade (Ref = Grade1)		<0.001		<0.001		0.002		0.376		0.069
Grade 2	1.30 [1.05–1.59]	0.014	1.52 [1.14–2.03]	0.004	1.47 [1.07–2.03]	0.018	1.08 [0.49–2.36]	0.854	0.35 [0.08–1.46]	0.149
Grade 3	1.96 [1.57–2.44]	<0.001	2.14 [1.50–3.05]	<0.001	2.04 [1.38–3.02]	<0.001	1.80 [0.67–4.86]	0.244	0.25 [0.05–1.33]	0.105
pN (Ref = pN0)		<0.001		<0.001		<0.001		0.076		<0.001
pN1a	1.34 [1.13–1.58]	0.001	1.27 [0.96–1.67]	0.089	1.32 [0.96–1.80]	0.083	1.22 [0.64–2.33]	0.55	1.43 [0.70–2.94]	0.326
pN1mi	1.35 [0.90–2.02]	0.145	1.64 [0.97–2.78]	0.066	1.58 [0.87–2.85]	0.130	2.57 [0.79–8.37]	0.117	4.04 [1.23–13.28]	0.022
pN2a	2.07 [1.69–2.54]	<0.001	1.42 [0.97–2.08]	0.074	1.88 [1.22–2.90]	0.004	1.51 [0.69–3.30]	0.304	1.49 [0.63–3.51]	0.363
pN3a	2.28 [1.80–2.88]	<0.001	2.67 [1.74–4.11]	<0.001	2.93 [1.77–4.86]	<0.001	2.81 [1.25–6.3]	0.012	4.61 [2.11–10.07]	<0.001
pNX	4.77 [3.24–7.02]	<0.001	16.27 [7.92–33.41]	<0.001	24.30 [8.94–65.90]	<0.001	9.92 [2.2–44.65]	0.003	86.4 [15.38–485.31]	<0.001
missing	2.54 [1.90–3.38]	<0.001	3.00 [1.77–5.08]	<0.001	3.06 [1.65–5.67]	<0.001	1.2 [0.16–9.17]	0.857	2.14 × 10^−7^ [0–∞]	0.996

**Table 4 cancers-13-03799-t004:** iAUC and concordance of the API (Cox regression), the survival tree model and the NPI for the test sets of the different cohorts.

	Total Cohort-6654		Filtered Cohort-3744		Subtype NST (WHO 8500/3)		All Special Types (Not WHO 8500/3)		Subtype Invasive Lobular (WHO 8520/3)	
	iAUC	Conc	iAUC	Conc	iAUC	Conc	iAUC	Conc	iAUC	Conc
API	0.710	0.708	0.671	0.672	0.689	0.699	0.601	0.566	0.415	0.413
Tree	0.720	0.704	0.656	0.650	0.642	0.645	0.510	0.514	0.504	0.508
NPI	0.639	0.668	0.646	0.652	0.664	0.674	0.587	0.563	0.545	0.542

**Table 5 cancers-13-03799-t005:** Comparison of the iAUCs of the API, the survival tree model and the NPI for the test sets of the different cohort using Wilcoxon rank sum tests for dependent samples.

	Total Cohort-6654	Filtered Cohort-3744	Subtype NST (WHO 8500/3)	All Special Types (Not WHO 8500/3)	Subtype Invasive Lobular (WHO 8520/3)
API vs. Tree	0.529	<0.001	<0.001	<0.001	1.000
API vs. NPI	<0.001	<0.001	<0.001	0.129	1.000
Tree vs. NPI	<0.001	<0.001	1	1.000	0.789

## Data Availability

Due to ethical concerns, supporting data cannot be made openly available.

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
