# Peer review of "Altona Prognostic Index: A New Prognostic Index for ER-Positive and Her2-Negative Breast Cancer of No Special Type"

_cancers, 2021, doi:10.3390/cancers13153799_

Round 1

Reviewer 1 Report

This article investigated the validity of the Nottingham prognostic index (NPI) and newly developed Altona prognostic index (API) in ER-positive and Her2-negative breast cancer of no special type. And authors concluded API was significant superior to the NPI in those patients.

This study is novel and interesting. And it is well organized systematically and logically.

However, as a limitation of this study, it is doubtful how many API will be used in real world in the era when multigene assay is being implemented as a prognostic factor. In addition, this study was retrospectively performed in a single center with a small number of the patients.

Despite these limitations, I think it is a study that can help readers in a way.

Therefore, I ask that authors should revise it based on the below commentary.

(1) If you tried to develop a new prognostic index, why didn't you consider other factors (ex, Ki67 etc...) than age?

(2) In the table 1, the sum of each group and the total sum of histologic grade 2, ER negative, pT1b, N missing and HER2 negative do not match. Please revise it.

(3) In the manuscript, there is no explanation for table 2. Please add it.

(4) Please correct the keywords to MESH term.

Reviewer 2 Report

The manuscript provides a comprehensive overview of the development and validation of a prognostic tool that incorporates age to predict breast cancer prognosis. The authors are to be commended for the systematic and logical organization of the manuscript. The methods are clear and robust. However, there were few issues concerning the overall objective of the study, measures and certain key design decisions as noted in the specific comments below.

Introduction and Objectives-

The prognostic value of age, grade and receptor-based subtype have been well known, and the AJCC 8th edition has incorporated grade and receptors status into the staging system. As written, it is not clear to me what was the exact purpose of the current study? If the primary study purpose was to develop a prediction model for breast-cancer specific survival, please make it clear and describe whether there are already existing prediction models for prognosis of stage I-III breast cancer patients, and how this new prediction model is going to be different from the existing prediction models, and how it is expected to be used in clinical practice?

Currently there are at least eight online clinical decision tools that incorporate clinical characteristics to predict breast cancer outcomes including breast cancer death and recurrence (please see the list below). It is not clear why it was necessary to develop another prognostic model and a tool to provide survival estimates. What is the overall contribution of this study to prognostication in breast cancer? Did the authors use the existing models to inform the development of their model and tool?

1. CancerMath Breast Cancer Outcome Calculator. Laboratory for Quantitative Medicine, Massachusetts General Hospital. http://www.lifemath.net/cancer/breastcancer/outcome/index.php.
2. Cleveland Clinic Risk Calculator Library. Department of Quantitative Health Sciences, Cleveland Clinic Lerner Research Institute. http://riskcalc.org:3838/.
3. Breast Cancer Nomograms Prediction Tool. Memorial Sloan-Kettering Cancer Center. http://nomograms.mskcc.org/breast/index.aspx. Accessed August 24, 2018.
4. Ravdin PM, Siminoff LA, Davis GJ, et al. Computer program to assist in making decisions about adjuvant therapy for women with early breast cancer. Journal of clinical oncology: official journal of the American Society of Clinical Oncology. 2001;19(4):980-991.
5. Predict. National Health Service. http://www.predict.nhs.uk/index.html.
6. Clinical Care Options. Clinical Care Options Web Site. https://www.clinicaloptions.com/.

7. Clinical Calculators. MD Anderson Cancer Center. https://www.mdanderson.org/for-physicians/clinical-tools-resources/clinical-calculators.html.
8. IBTR! Version 2.0 Breast Cancer Model. Tufts Medical Center. https://www.tuftsmedicalcenter.org/ibtr/.

2) Why wasn’t race/ethnicity or level of comorbidity included in the model? These have been shown to be important predictors of breast cancer survival.

3) Was it necessary to limit the sample to women who received surgery as the initial treatment? Moreover, did the researchers consider the effect of type of surgery?

 4) Why wasn’t tumor size considered separately in the model to provide predictions based on finer levels of tumor size?

Results-
The steps involved in the re-calibration to improve the predictions in the validation data set are not clear.

Discussion-
It is not clear how the statistical method used in the current study compares to other prediction models and statistical approaches. What are the strengths, limitations and other applications of this approach?

There is no discussion about the relative performance of their model compared to existing models in the literature. The readers would benefit from learning how the model compares to existing models that incorporate more determinants such as surgery status, race/ethnicity, margin status, treatment etc.

Round 2

Reviewer 1 Report

Thank you for your efforts.

It was revised well according to the review opinion.

But, please check the review opinion (2) again.

(In the table 1, the sum of each group and the total sum of histologic grade 2, ER negative, pT1b, N missing and HER2 negative do not match. Please revise it.)

I meant that the overall number (n=6654) and the sum of each group (M 8500/3, M8520/3, ...) of do not match.

Thank you.

Author Response

Point 2: In the table 1, the sum of each group and the total sum of histologic grade 2, ER negative, pT1b, N missing and HER2 negative do not match. Please revise it.

I meant that the overall number (n=6654) and the sum of each group (M 8500/3, M8520/3, ...) of do not match.

Response 2: The numbers of the different WHO-groups were adapted, so that the sum of all subtypes is correct now (see Table 1 of the revised mansucript). Thank you.

Reviewer 2 Report

Thank you for your responses

I think there is a sentence missing for point 1, "If our manuscript of the API is accepted and published, we will provide the index formula on the homepages of our department a"

Author Response

Addition Point 1 (missing sentence): 

We apologize for the mistake. The incomplete sentence for response 1 should have been:
If our manuscript of the API is accepted and published, we will provide the index formula on the homepages of our own department and those of the collaborating breast centers for online public access.